# Making Cloud Spot Instance Interruption Events Visible

## ABSTRACT

Public cloud computing vendors offer a surplus of computing resources at a cheaper price with a service of spot instance. Despite the possible great cost savings from using spot instances, sudden resource interruption can happen as resource demand changes. To help users estimate cost savings and the possibility of interruption when using spot instances, vendors provide diverse datasets. However, the effectiveness of using the datasets is not quantitatively evaluated yet, and many users still rely on hunch when choosing spot instances. To help users lower the chance of interruption of the spot instance for reliable usage, in this paper, we thoroughly analyze various datasets of the spot instance and present the feasibility for value prediction. Then, to measure how the public datasets reflect real-world spot instance interruption events, we conduct real-world experiments for spot instances of AWS, Azure, and Google Cloud. Combining the dataset analysis, prediction, and real-world spot instance interruption experiment result, we show the feasibility for lowering the possibility of interruption events significantly.

## CCS CONCEPTS

• **Computer systems organization** → **Cloud computing**; • **Information systems** → **Web log analysis**.

## KEYWORDS

cloud computing, spot instance, interruption modeling, enhancing reliability, spot instance datasets

### ACM Reference Format:

Anonymous Author(s). 2018. Making Cloud Spot Instance Interruption Events Visible. In *Proceedings of Make sure to enter the correct conference title from your rights confirmation emai (Conference acronym 'XX)*. ACM, New York, NY, USA, 12 pages. https://doi.org/XXXXXXX.XXXXXXX

## 1 INTRODUCTION

The cloud computing provides compute resources elastically without the burden of system operation overhead and changes the way we consume compute resources. The core success of the cloud computing is owing to its on-demand billing model that allows users to dynamically start and stop instances based on its application needs and pay for what they have actually used. To support elastic resource usage, cloud service providers should prepare abundant resources to meet resource usage spikes. Such a plentiful resource preparation inevitably results in resource wastage when computing demand is low. To encourage instance usage when demand is

low, public cloud vendors provide the surplus computing resources at a discounted price which can be more than 90% cheaper than the on-demand price in some cases. The billing model is called spot instances, and they are offered by most public cloud service providers, AWS [1], Azure [52], Google Cloud Platform (GCP) [19], Alibaba [34], Oracle Cloud Preemptible Instances, and IBM Transient Virtual Servers. Due to the cost efficiency of spot instances, they are widely used for diverse applications, such as web server, data processing, and parallel processing batch jobs.

In the early days of spot instance offering, most vendors adopted an auction mechanism where a user sets a bidding price, and a provider sets a spot price. If the bidding price is higher than the spot price, a user gets granted an instance and pays for the spot price, not the bidding price. Based on the compute resource demands, a service provider changes the spot price, and if the spot price becomes higher than the initial bidding price, a spot instances is revoked, which is generally referred to as instance interruption. Such an instance interruption can negatively impact applications' reliability, and users should be prepared for the event. To help users enhance reliability when using spot instances, cloud vendors provide diverse datasets publicly through web. One of the most representative datasets is the spot instance price that presents the spot price at a specific time. Many research was conducted using the spot instance price dataset to analyze the spot price itself [1, 14, 22, 34, 49], using the spot instance dataset to enhance reliability for diverse applications [2, 12, 20, 28, 32, 41, 43, 47], or suggesting an optimal bidding price using the spot price prediction [3, 16, 26, 27, 42, 58].

The spot price dataset was a precious source of information when using spot instances reliably especially for AWS, but it changes with the new spot instance operation policy. With the new policy, the advertised spot price does not reflect the spot instance interruption anymore, and the frequency of the data change becomes very low [6, 21] which greatly degrades the application of the dataset and invalidates the outcome from the prior research. Meanwhile, new data sets of spot instances are released on the web to help users build a reliable resource pool of spot instances [7]. The new datasets include the interruption ratio over the prior month dataset provided by Azure and AWS, and the instant availability information provided by AWS. The new datasets can be of great help for spot instance users as they provide the historical interruption ratio and instantaneous availability information. However, thorough investigation of the newly provided datasets has not been conducted yet, and quantitative evaluation about the correlation of the dataset and the spot instance interruption has not been carried out yet. Given that sudden spot instance interruption is a major hurdle when adopting a spot instance as a main compute resource pool, it is crucial to estimate the likelihood of a spot instance to be interrupted at least in the near future. However, predicting future spot instance interruptions or building a statistical model for interruption behavior can be very challenging from outside of cloud service vendors' view.

To quantitatively evaluate and model spot instance interruption events, one should know when the interruption events happen for diverse instance types that are located in global regions and availability zones. The service providers may have such records, but the information is not publicly available. For example, Yang et. al. [52] proposed a spot instance interruption prediction model for Azure spot instances by using the internal dataset of interruption records that are not publicly available, and it limits the development of publicly available research outcomes. To model spot instance interruption events without proprietary information, SciSpot [24] and Pham et. al. [37] conducted real-world experiments to observe spot instance interruption events by simply running spot instances until an interruption happens for GCP and AWS, respectively. Even after gathering the interruption event experiment results, without knowing the internal mechanism of spot instance operations, it can be challenging to select features that really impact the interruption events. In case of AWS spot instances, the spot price was a good feature for the interruption prediction, but it is not valid anymore after the operation policy change [6, 21].

Based on the observation that the prediction and modeling of spot instance interruption events is challenging even with the newly released spot instance datasets available on the web, we first analyze the characteristics of the spot price, prior period interruption ratio, and instant availability dataset that are provided by AWS, Azure, and GCP where applicable. To validate whether the public datasets reflect the real behaviors of spot instance interruptions and compare spot instance reliability of multiple vendors, we conducted spot instance interruption tests for AWS, Azure, and GCP. Based on the public dataset analysis and real-world spot instance interruption experiment result, we argue that precise prediction of instant spot instance availability dataset value can lower the probability of spot instance interruption and confirm the argument quantitatively. Overall, we conclude that spot instances offered by Azure showed the highest reliability followed by GCP and AWS. Though AWS showed the lowest reliability, we find out that the dataset provided by AWS is really helpful to conjecture spot instance reliability. By using a proposed dataset value predictor, the spot instance running time of AWS can increase by 63.2% for initially high score instances and by 168% for instances with initially low score.

In summary, major contributions of this paper is as follow.

- Comparing spot instance reliability of AWS, Azure, and GCP. To the authors' best knowledge, this is the first work to compare the spot instance reliability of multiple vendors.
- Thorough analysis of the new spot instance datasets including price, interruption ratio, and availability, to discover correlations with the spot instance interruptions
- Proposing a spot instance dataset value prediction model to enhance spot instance reliability
- Proposing a general guidance when using spot instances in a multi-cloud environment

Section 2 discusses publicly available spot instance dataset, and Section 3 analyzes various spot instance datasets provided by AWS, Azure, and GCP. Section 4 proposes a model to predict instant availability dataset and evaluates the model quality. Section 5 evaluates

how the spot instance dataset distribution is correlated to real interruption behavior and presents the practicality of dataset value prediction to enhance spot instance reliability.

## 2 CLOUD SPOT INSTANCE AND DATASETS

When using spot instances, the cost saving ratio over the on-demand instance and the reliability of a spot instance are the most important metric for most users. To help users estimate the benefit and risks when using spot instances, the cloud vendors provide various datasets; the spot instance price dataset from the beginning and the more recent interruption ratio and instant availability information dataset.

### 2.1 Depreciation of Price Dataset

Since the introduction of the spot instance service in 2009, the spot price dataset is publicly available in the web and triggered many research from various perspectives. Statistical analysis of the spot instance price dataset was performed in a comprehensive way [1, 14, 16, 22, 31, 34, 38, 49, 50]. Other works focused on proposing an optimal bidding price to reduce the risk of instance interruption [3, 18, 27, 30, 40, 42, 44, 55, 58]. Another type of work focused on running various applications on spot instances reliably using spot instance price datasets, such as web server [2], big data processing [51], deep learning training [28, 46] and inference [57], batch processing [32, 43], and scientific high-performance computing applications [59]. Other works are carried out using a dataset from other vendors, Azure [52, 53], GCP [19, 24], and Alibaba cloud [34].

One of the reasons that many research was conducted using AWS spot instance and its price dataset is owing to its dynamically changing price patterns, which reflect the spot instance interruption events very well. By comparing the advertised spot price and bidding price, a user can easily conjecture the interruption possibility. However, modeling spot instance interruption using a price dataset becomes impossible since the spot instance operation policy change [6, 21]. With the new change, the spot price does not change as often as before. More importantly, the spot price does not indicate an instance interruption; though the advertised spot price is lower than a bidding price, an interruption can still happen. The operation policy change makes the AWS spot instances similar to other vendors' spot instances which implies the spot price rarely changes, and it makes most of the previous research that relied on the spot price datasets becomes useless.

### 2.2 Appreciation of Availability Dataset

As the usefulness of spot price dataset decreases from the perspective of inferring spot instance reliability, the service vendors started to provide new types of datasets related to instance availability. AWS and Azure provide the ratio of interruption of an instance in the prior time frame, e.g., 30 days. The dataset classifies interruption ratios to five categories, less than 5%, between 5% and 10%, between 10% and 15%, between 15% and 20%, and more than 20%. By using the dataset of previous interruption rate, users are expected to infer spot instance's future reliability.

Different from the interruption ratio dataset which simply provides the statistics from the previous period, AWS provides a new data set called Spot Placement Score (SPS). The vendor did not

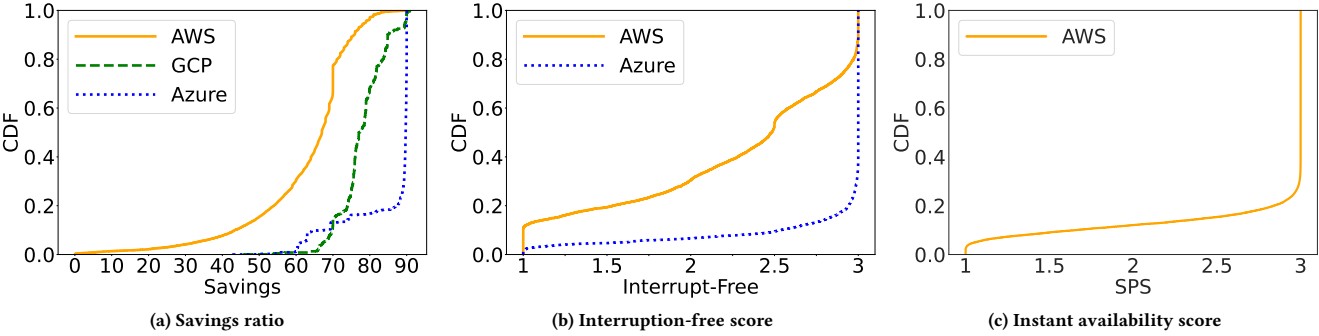

(a) Savings ratio  (b) Interruption-free score  (c) Instant availability score

**Figure 1: Spot instance dataset value distribution from multiple vendors**

disclose the internal detail of how the score is calculated, but it is known to present a timely spot instance's availability. In the SPS, a type of spot instance is assigned an integer score ranging from one to three; the higher score implies more availability.

## 3 SPOT INSTANCE DATASET CHARACTERIZATION

To better understand characteristics of different spot instance datasets, we present empirical analysis result of publicly available spot instance datasets which can be accessed from the web. We could access the spot price dataset of AWS, Azure, and GCP. For the spot instance interruption ratio dataset, we get datasets from AWS and Azure. For instant availability information, we get the AWS SPS dataset. In the analysis, datasets from 1 November 2022 to 31 August 2023 were used.

Figure 1 shows the Cumulative Distribution Function (CDF) of spot instance dataset. Figure 2a compares the savings ratio, which is calculated as $(1.0 - \frac{SpotPrice}{On-demandPrice}) \times 100$, of AWS, Azure, and GCP. Figure 2b compare the the prior period's instance interruption ratio of AWS and Azure. Both vendors provide the dataset with values in the range of five categories, and we match each category to a numeric value between 1.0 and 3.0 increments by 0.5. The most frequent interruption ratio of more than 20% is matched to 1.0, and the least value of less than 5% is matched to 3.0. We name the converted value as *interrupt-free score*. Figure 2c presents the distribution of the instant availability dataset provided by AWS. We use the SPS dataset with the provided score range. To distinguish vendors, we use yellow-solid, blue-dotted, and green-dashed lines to indicate AWS, Azure, and GCP, respectively. In the figures, the vertical axis shows the distribution, and the horizontal axis shows the score. For all three sub-figures, the large score means the better and more reliable status.

In the figures, it is apparent that AWS has the least cost savings (Figure 2a) and a higher spot instance interruption ratio (Figure 2b). The GCP provides only the price dataset, and it shows higher median cost savings ratio than AWS. Regarding the interrupt-free score, the median score of AWS is around at 2.5 which means the interruption ratio between 5% and 10%, while that of Azure is around at 3.0, which is less than 5%. Only AWS provides the instant availability

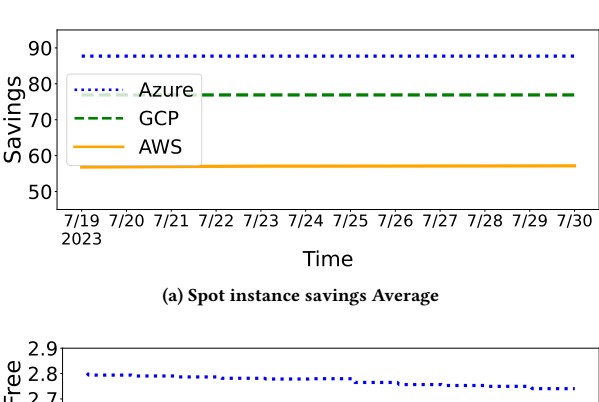

(a) Spot instance savings Average

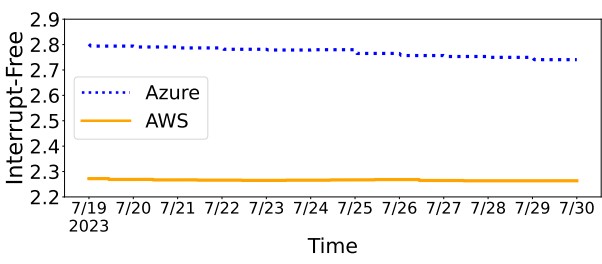

(b) Interruption-Free Score Average

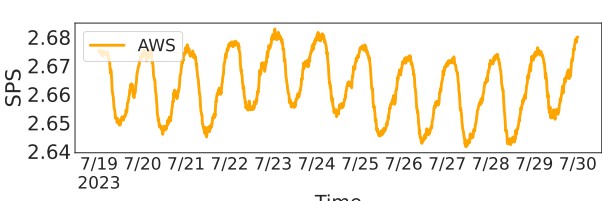

(c) Instant Availability Score Average

**Figure 2: Spot datasets that have been changed during the observation period**

information, and we can observe that most of spot instances have been allocated near 3.0 point which is the best score.

Figure 2 shows the temporal change pattern of three spot datasets averaged for all gathered instance types in each sub-figure. Figure 2a compare the spot instance savings ratio, and we can observe that regardless of the vendors, savings value rarely changes over time.

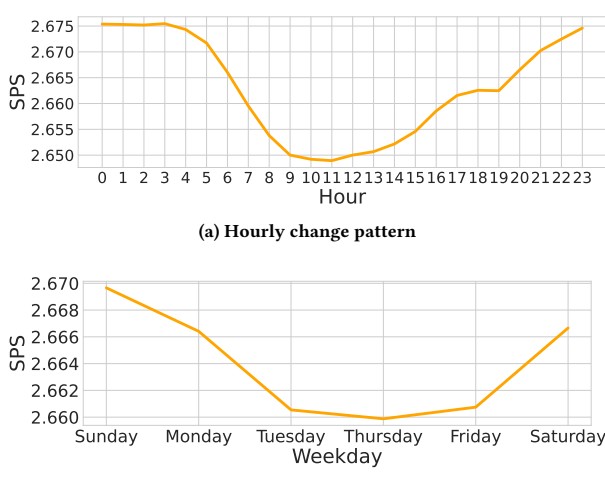

**(a) Hourly change pattern**

**(b) Daily change pattern**

**Figure 3: The change pattern of instant availability dataset shows a strong hourly and daily change pattern**

The low frequency of price change of AWS spot instance concurs to the prior work that analyzed the recent AWS spot instance behavior change [6, 21]. Figure 2b presents the temporal change pattern of interrupt-free score. Similar to cost savings, both AWS and Azure do not show noticeable change over time. Figure 2c presents the temporal change pattern of AWS SPS dataset. Different from the cost savings and interruption dataset, the SPS dataset shows a sinusoidal periodic pattern for different days. From the analysis, we can uncover the unique and noticeable change pattern of SPS that intrigues further research and analysis. Please note that the range of vertical axis is different for distinct dataset. We closely observed the pattern of savings and interruption-free score in a small scale but could not find a pattern similar to SPS.

To further analyze the pattern of SPS value change, Figure 3a shows SPS values averaged over an hour of a day. In the analysis, we used globally located resources in multiple regions and used the local time zone of each availability zone. From the figure, we can discover that the SPS is lower in the morning around at 9, 10, and 11. Then, the SPS gradually increases and reaches its maximum at night which is the time that cloud usage is expected to be low. To see if the SPS value change has a weekly pattern, Figure 3b presents the average SPS value by day which is shown in the horizontal axis. We can observe that the SPS value is higher on weekends and lower on week days. Please note that we analyzed the spot price and interruption ratio similarly, but we could not observe a noticeable pattern.

We could observe more frequent and periodic data update pattern of AWS SPS dataset. To further analyze how often spot instance dataset changes, Figure 4 shows the CDF of data change frequency. The vertical axis shows the distribution, and the horizontal axis shows how long a value remains unchanged. The larger value, which is located at the right side of the figure, implies that a value does not change for a longer period of time. In the figure, the distribution of all the datasets are presented. Among them, AWS

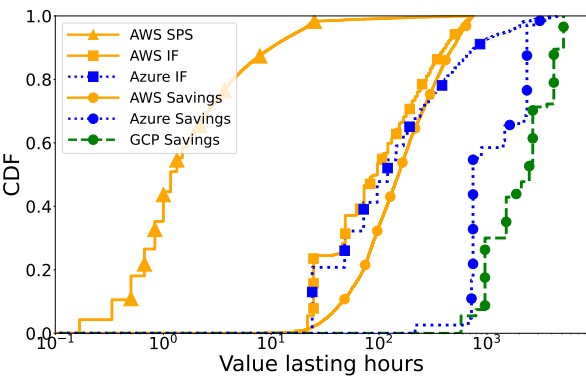

**Figure 4: The CDF of value change frequency. The SPS dataset provided AWS shows the most frequent data update**

SPS with a yellow solid line with triangle markers shows the most frequent update compared to other dataset. For cost savings dataset, we assume that a cost saving value has changed when the ratio has increased or decreased more than 3%.

From the various spot instance dataset analysis, we could uncover that spot instances from **Azure and GCP show relatively higher cost savings than AWS**. The **interruption ratio of AWS is higher than that of Azure**. Only the **AWS SPS dataset shows a pattern of daily and hourly change**. The values of **AWS SPS changed frequently**, and the change frequency of other datasets are longer than a day.

## 4 PREDICTING SPOT INSTANCE DATASET CHANGE

When using spot instances, the prediction of future interruption events and probability estimation can be of great help to improve reliability. To achieve the goal, it is important to discover appropriate features that correlate with the interruption events. The analysis of the spot instance dataset reveals that the price and interruption ratio information has little change over time, while the instant availability information of AWS SPS changes often and regularly. Considering the recent research outcomes about spot instance price [6, 21, 29], the spot price information has little correlation with the spot instance interruption. The spot instance interruption ratio presents the prior 30 days of interruption statistics, and it can be far from predicting the future availability. In this context, we propose a model to predict the instant availability dataset expecting that the precise prediction can help to improve spot instance reliability, which will be evaluated later.

### 4.1 Modeling Instant Availability Dataset

The instant availability dataset prediction model uses the prior period's SPS as input features and predict future SPS values. Formally speaking, the input dataset and features of $X$ are composed of $N$ distinct instance types in different availability zones, which is the unit of the distinct SPS being provided, and $D$ features which represent an SPS value at different timestamp. Assuming that data collection of SPS dataset occurs with a period of $p$ minutes, we can use the previous $D$ collected dataset for training, which means that we use

|  | 3H | 6H | 12H | 24H | 48H | 72H |
|---|---|---|---|---|---|---|
| LR | 0.96 | 0.97 | 0.98 | 0.97 | 0.97 | 0.96 |
| RF | 0.96 | 0.98 | 0.98 | 0.98 | 0.97 | 0.97 |
| XGboost | 0.97 | 0.99 | 0.98 | 0.98 | 0.97 | 0.97 |
| SVC | 0.96 | 0.97 | 0.97 | 0.98 | 0.96 | 0.96 |
| Prophet | 0.96 | 0.96 | 0.97 | 0.98 | 0.97 | 0.97 |
| ARIMA | 0.95 | 0.95 | 0.97 | 0.97 | 0.96 | 0.96 |

**Table 1: F1-score of various models for SPS prediction**

datasets collected in the last $D \times p$ minutes. Similarly, the target dataset to predict is noted as $Y$ that has $N$ distinct instance types and next $K$ SPS values to predict. $x_{id}$, where $i = 1 : N, d = 1 : D$, means the $i$-th instance type and $d$-th index SPS value, and $y_{ik}$, $i = 1 : N, k = 1 : K$, means the $i$-th instance type and $k$-th index SPS. The train dataset is defined as follow.

$$\mathcal{D} \triangleq \{(x_{id} \rightarrow y_{ik})|(x_{id}, y_{ik}) \in \{1, 2, 3\}\} \quad (1)$$

Each value of $x_{id}$ and $y_{ik}$ takes one of $\{1, 2, 3\}$ that is a possible SPS value. In defining training dataset, we need to decide how many previous measured datasets to use, which is $D$, and how many future dataset to predict, which is $K$. Our thorough empirical analysis reveals that the selection of $D$ and $K$ do not have much impact to the overall prediction accuracy (Figure 5).

## 4.2 Accuracy of Availability Dataset Predictor

Using the train dataset, $\mathcal{D}$, in equation 1, we apply diverse classification models of Linear Regressor (LR) [35], Random Forest (RF) classifier [9], XGBoost [11], Support Vector Classifier (SVC) [5], Prophet [45], and ARIMA [8]. Table 1 compares the F1 score for different prediction models to evaluate both precision and recall [39]. In the train, we used the SPS dataset from 16 July 2023 to 31 July 2023. For testing, we used the dataset from 1 August 2023 to 6 August 2023 which is exclusive to train dataset period. We divide the test dataset in half and used the prior half period as the model's input. Of the later half dataset, we variate the prediction period, $k$, to the next 3, 6, 12, 24, 48, 72 hours, and they are shown in the columns.

We can discover that most models show decent prediction quality. Among them, the XGBoost shows the best result, and we use them in the following descriptions.

Figure 5 shows the heatmap of F1-score of a SPS prediction model built by using XGBoost with respect to different input time window (the horizontal axis) and the different output time window (the vertical axis) presented in hours. In the figure, lighter region implies higher F1-score. As we can see from the figure, regardless of the input and output time window, the F1-score shows consistently high scores except when predicting further future (72$H$) using only recent values (3$H$).

Figure 6 presents the feature importance which is extracted from a built XGBoost model. The vertical axis shows the importance of each feature, and the horizontal axis shows the index of time-series feature. The lower index number implies more recent dataset. As shown in the figure, recent values dominate the feature importance, where till 24 hours takes 79% of total importance.

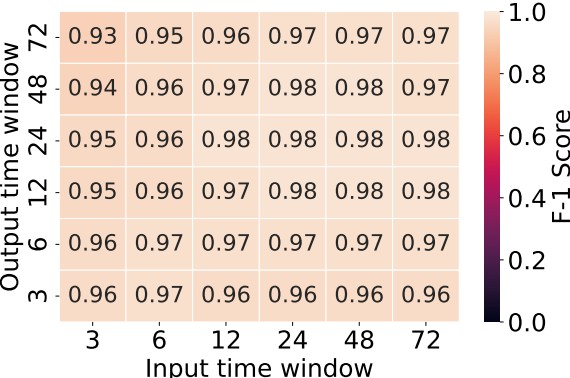

**Figure 5: Heatmap of F1-score for XGBoost for modeling with respect to different input and prediction output time window**

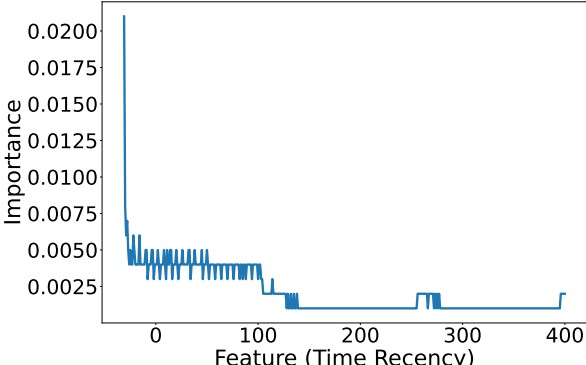

**Figure 6: The importance of features when building a SPS prediction model**

## 5 EVALUATION

This section evaluates the effectiveness of a heuristic to improve the reliability of the spot instance through a detailed analysis of spot instance interruption events. More specifically, we would like to answer the following research questions.

**RQ-1** For spot instances offered by major public vendors, which vendor provides the most reliability with respect to the interruption?

**RQ-2** Of many spot instance datasets, which dataset is most correlated with spot instance interruption?

**RQ-3** Can the prediction of SPS value lower the chance of spot instance interruption?

## 5.1 Spot Instance Interrupt Analysis

We first compare the interruption behaviors of multiple cloud vendors (**RQ-1**). To examine interruption events, we conducted real-world experiments by running a spot instance until an interruption event happens. We choose 875 distinct instance types in AWS, Azure, and GCP located globally. When choosing arbitrary instance types, we try to distribute evenly for different values in the cost savings, interruption ratio, and instant availability datasets. The

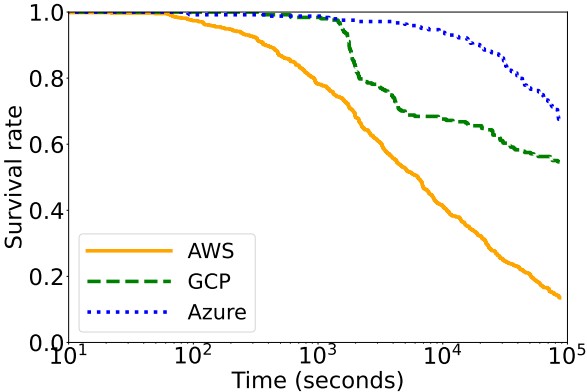

**Figure 7: Kaplan-Meier Estimator analysis to infer the probability of spot instance survival rate. Azure shows the best reliability followed by GCP and AWS.**

experiments had been conducted for 24 hours for each instance types. When an interruption event happens, we mark the event and keep requesting the spot instance until it fulfills again.

To quantitatively compare the survival rate of the spot instances, the probability that a spot instance remains running, we applied the Kaplan-Meier Estimator [25] which is a non-parametric statistic to estimate the survival rate of a lifetime dataset. It is generally used in the hospital environment to measure the fraction of people who live after treatment or the length of time people remain unemployed after losing a job [33]. Based on the general usage of the statistic, applying it to model spot instance survival probability is well suited. Kaplan-Meier Estimator is calculated as follows.

$$\widehat{S}(t) = \prod_{i:\ t_i \le t} \left( \frac{n_i - d_i}{n_i} \right)$$

The survivor function ($\widehat{S}(t)$) at time $t$ is defined as the probability a life time (spot instance running time) is longer than $t$. $n_i$ is the number of survived individuals (running spot instances) until time $t_i$, and $d_i$ means the number of deaths (spot instance interruption) at time $t_i$.

Figure 7 compares $\widehat{S}(t)$ of AWS, GCP, and Azure. In the figure, the horizontal axis shows the spot instance running time (survival time), and the vertical axis shows the distribution. The yellow-solid, green-dashed, and blue-dotted lines present the distribution of AWS, GCP, and Azure respectively. From the figure, it is evident that the AWS shows the lowest survival rate followed by GCP and Azure. The median running time of AWS spot instance is 1.2 hours, and over the half of GCP and Azure instance did not experience interruption during the 24 hour experiments. The shortest $P90$ running time of AWS, GCP, and Azure is 0.02, 0.5, and 5 hours, respectively.

Next, we assess the correlation of various spot instance datasets and the interruption events (**RQ-2**). Figure 8 compares Kaplan-Meier estimator distribution of AWS experiment result grouped by different spot instance dataset. Figure 8a compares survival time distribution with respect to different interrupt-free score, and Figure 8b compares with respect to SPS scores. For both datasets,

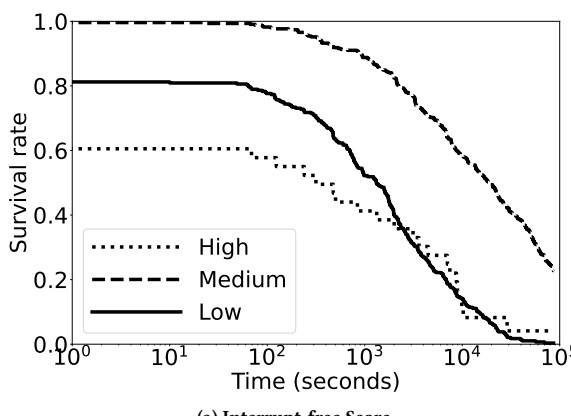

**(a) Interrupt-free Score**

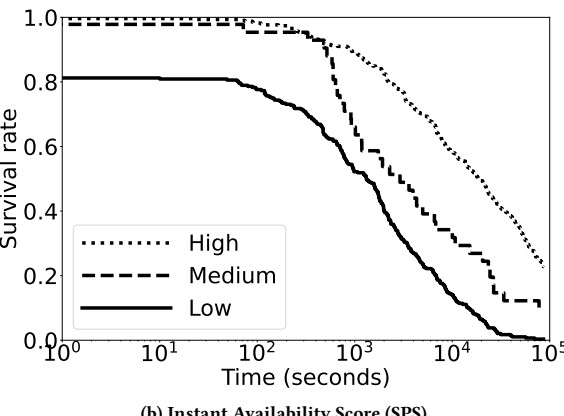

**(b) Instant Availability Score (SPS)**

**Figure 8: Kaplan-Meier Estimator to compare the probability of spot instance survival rate with respect to different spot instance datasets**

the higher score implies the higher availability. The scores of *Low* (1.0), *Medium* (2.0), and *High* (3.0) are presented with solid, dashed, and dotted lines, respectively. From both figures, we can observe that higher score values shows higher survival rate than lower score values. We omit the figures of savings for AWS, Azure, GCP, and interrupt-free score for Azure because they do not show a noticeable pattern as AWS interrupt-free score and SPS do.

So far, we have analyzed the spot instance survival time. Next, we will analyze using a different metric. Figure 9 compares the elapsed time between a spot instance interruption start and the node becomes fulfilled again. The shorter time implies that a spot instance is becomes available again shortly after an interruption happens which means a higher availability. Sub-figures group instances according to the spot instance dataset. In each figure, the yellow-solid, blue-dotted, and green-dashed lines indicate the result from AWS, Azure, and GCP, respectively. In each line, we mark a symbol of △ for *High* and ▽ for *Low* value after categorization.

The cost savings ratio dataset (Figure 9a) does not show a noticeable difference among *High* and *Low* values for all AWS, Azure, and GCP. GCP and Azure show a similar latency distribution, while that

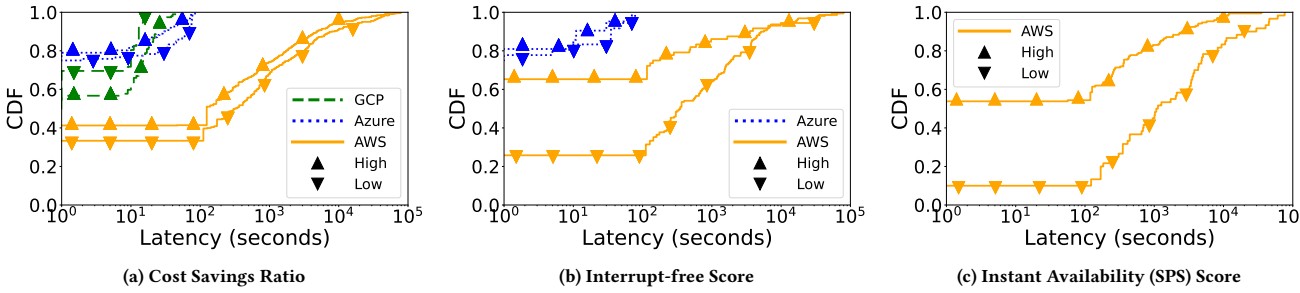

(a) Cost Savings Ratio                          (b) Interrupt-free Score                          (c) Instant Availability (SPS) Score

**Figure 9: A CDF of latency until a request becomes fulfilled for AWS, Azure, and GCP spot instances categorized by different dataset values of High (△) and Low (▽). The lower latency implies the higher availability.**

of AWS shows much less availability than the others. Regarding the interrupt-free score (Figure 9b), Azure did not show a different pattern between *High* and *Low* values, but AWS shows a distinct pattern that follows the advertised score. The high interrupt-free score of AWS spot instances show faster fulfillment time after an interruption. The instant availability data (Figure 9c) shows a noticeable difference between *High* and *Low* that the higher SPS value shows much lower latency for a fulfillment after an interruption (higher availability), and it follows the advertised score characteristic very well.

## 5.2 Effectiveness of Instant Availability Dataset Prediction

We have discovered that the spot instance instant availability dataset provided by AWS SPS dataset is beneficial to model spot instance interruption events and availability. To go one step further and answer **RQ-3** of whether predicting future SPS value can help to increase to predict chances of spot instance interruption, we compare the distribution of spot instance interruption with different values of SPS prediction in Figure 10. The vertical axis shows distribution, and the horizontal axis presents the spot instance running (survival) time. Figure 10a presents the life time of spot instances whose SPS value is high (3.0) when a spot instance request is made. The SPS value can change over the course of experiment, and we categorize the predicted SPS values in a bin size of 0.5. The dotted line shows when the average predicted SPS value is 3.0, which means SPS value is expected to remain constant at the high value. A dashed-dotted line indicates when the expected SPS average is higher than 2.5 but lower than 3.0. The other lines are expressed in a similar way. From the figure, we can discover that despite the same initial high SPS value of 3.0, ignorant of future SPS value can significantly hurt the reliability of a spot instance. For instance, the median spot instance survival time when the predicted SPS is high is 22 hours, but that of lower predicted SPS value is 16 hours.

Figure 10b shows the survival time of spot instances whose initial SPS value is low, 1.0. Similarly, the SPS can value can fluctuate, and we show the running time by grouping instances by the predicted SPS values. The solid line presents when the SPS value is expected to keep staying low at 1.0, and it shows very short running time. Otherwise, even if the initial SPS value is low, instances with a higher expected SPS values show higher survival time. For instance,

the median running time of spot instance whose SPS is expected to be consistently at 1.0 is 0 hours, while that of SPS being expected to be between 2.0 and 2.5 turns out to be about 8 hours. This finding is especially helpful when a user has a limited list of spot instance types, and their initial SPS is low. In that case, the prediction of SPS can greatly improve the reliability of the spot instance, though its current SPS is low.

In summary, we can answer research questions raised before as follow.

**RQ-1** For spot instances offered by major public vendors, which vendor provides the most reliability with respect to the interruption? **Answer** : The spot instance of Azure showed the most reliability followed by GCP and AWS.

**RQ-2** Of many spot instance datasets, which dataset is most correlated with spot instance interruption? **Answer** : AWS SPS dataset shows the highest correlations with the spot instance interruption. The cost savings and interrupt-free score do not show as high correlation as AWS SPS does.

**RQ-3** Can the prediction of SPS value lower the chance of spot instance interruption? **Answer** : Yes, the prediction of SPS dataset can definitely help to increase the spot instance reliability. When the initial SPS value is high, selecting an instance whose predicted value is consistently high can increase the expected spot instance running time by 63.2%. Even when the initial SPS value is low, selecting spot instances with higher expected SPS value can increase the expected spot instance running time by 168%

Overall, the spot instance datasets offered by Azure and GCP are not really useful when choosing appropriate instance types. However, the spot instance reliability of Azure and GCP is relatively high, and users may experience lower interruption regardless of instance selection. The spot instance offered by AWS shows the most interruptions. However, the spot instance datasets provided by AWS reflects the spot instance interruptions very well. Thus, different from Azure and GCP, when using a spot instance from AWS, users should be more cautious to choose appropriate instance type for reliable execution.

## 6 RELATED WORK

**Characterizing Spot Instance Dataset** : **Modeling and Using Spot Instance Dataset** : From the inception of the spot instance,

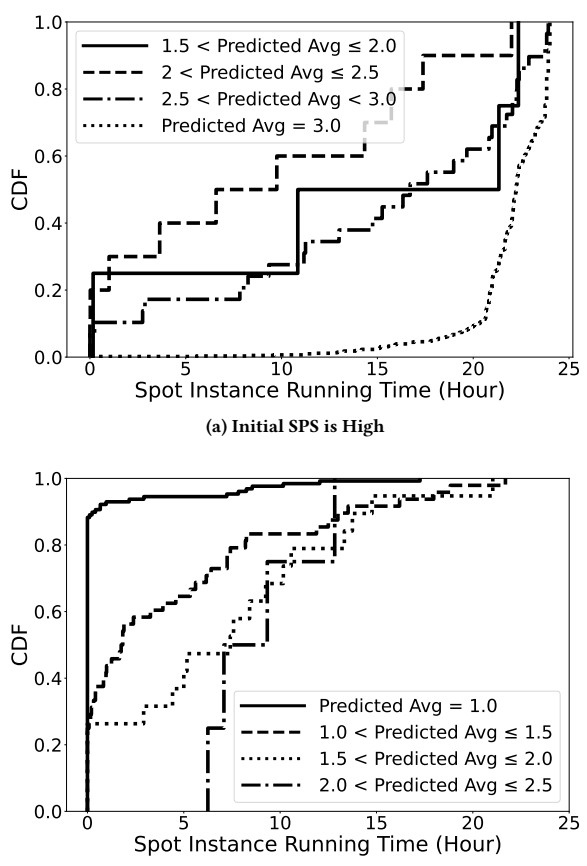

(a) Initial SPS is High

(b) Initial SPS is Low

**Figure 10: Distribution of spot instance availability when using only the current SPS value and predicted values**

the spot price dataset is widely used and analyzed to enhance reliability of spot instances. There were attempts to characterize the price dataset of the spot instance and relate the analysis to the spot instance interruption to decrease chances of interruption [14, 22, 31, 34, 38, 38, 49, 50]. Using the spot price and analysis result, Ali-Eldin et. al. proposed to deploy a web-server [2], Son et. al. proposed DeepSpotCloud [28] for DNN training tasks using GPU spot instances located globally. SeeSpotRun [12] for Hadoop [17] MapReduce [13], Flint [41] and Tr-Spark [51] for Apache Spark [56] are proposed for big-data processing. Online web services [2, 20], batch processing jobs [32, 43], and parallel processing of independent tasks [47] while mitigating straggler effect due to transient servers [4] are proposed. The aforementioned work relied on spot price dataset and the work becomes obsolete due to the spot instance operation change [6, 21]. The findings in this paper provide a new opportunity of enhanced spot instance reliability through instant availability dataset prediction without relying on spot price dataset. By selecting spot instances that are likely to be more available in the future by using the dataset prediction module, the application of spot instance can be further expanded.

**Spot Instance Price Prediction** : We showed that predicting the spot instance instant availability dataset is predictable using a machine learning classifier model with decent performance. Before this work, there are attempts to predict spot price value to predict future cost savings and interruption events. Khandelwal et. al [26] used the Random Forest, Fabra et. al [16] used a deep neural net model, and Alkharif et. al [3] used various time-series analysis statistical methods for price prediction. Due to the spot price operation policy change [6, 21], the previous work becomes obsolete, and new approaches, as this paper does, should be provided.

**Analyzing Spot Instance Interruption** : Compared to the analysis of the spot price dataset, the analysis and experiments of spot instance interruption and correlating it with the spot instance dataset were not conducted much. Pham et. al. [37] and Lee et. al. [29] conducted spot instance interruption experiments for only AWS instances to analyze the interruption pattern. For Azure, Yang et. al. [52] proposed a spot instance interruption prediction model based on a Transformer model by using internally available interruption trace of Azure. For GCP, Haugerud et. al. [19] and Kadupitiya et. al. [24] conducted interruption tests to model the behavior. To the best of the authors' knowledge, this paper is the first work to conduct spot instance interruption experiments for three major vendors of AWS, Azure, and GCP. Comparing the spot instance behavior can greatly help users to choose the most appropriate spot instances in a multi-cloud environment that is deemed to be widely adopted [10, 36, 54].

## 7 CONCLUSION

Cloud spot instances provide significant cost savings when using cloud resources with the risk of sudden instance termination. To help users better utilize spot instance, public service vendors provide diverse spot instance datasets, such as price, interruption ratio in the past period, and the instant availability information. Despite the diverse publicly available information, they are neither widely used nor analyzed except the spot price dataset which is really irrelevant to the spot instance interruption and reliability. To handle this issue, this paper thoroughly analyzes characteristics of various spot instance datasets and proposes a model to predict instant availability dataset to enhance reliability. To uncover the relationship of publicly available dataset and the spot instance interruption behavior, we conducted real-world experiments for the spot instance interruptions in the AWS, Azure, and GCP cloud. We discovered that the Azure spot instance shows the highest reliability, followed by GCP and AWS. Though AWS showed the worst reliability, we discovered that the instant availability dataset offered by AWS can be helpful to predict interruption events in the near future which was not possible by using datasets offered by Azure and GCP. Finally, by using the proposed instant availability score prediction, we showed that the median spot instance running time can improve by 63.2% for initially high score instances and by 168% for initially low score spot instances. We believe that our study will enable users to utilize cloud resources more efficiently, ensuring reduced costs and increased reliability, thereby optimizing their overall system performance.

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

## A FURTHER ANALYSIS OF SPOT INSTANCE DATASETS

The spot instance instant availability dataset, AWS SPS, provides quite significant insights about the service operation. To further analyze characteristics of the dataset, Figure 11 shows SPS value distribution grouped by different criteria. In the sub-figures, the horizontal axis shows the SPS value, and the vertical axis presents the distribution. Figure 11a presents the SPS value distribution with respect to the number of CPU cores. The number of cores are grouped into ones with more than 128 cores (solid line with triangle marker) while decreasing the core numbers in half. From the figure, it is evident that the SPS score is inversely proportional to the number of CPU cores. For instance, the median SPS of when there are more than 128 cores is 1.9, while that of one or two CPU cores is 3.0. This results concurs with the spot instance interrupt analysis experiments conducted for GCP ones [23, 53].

Figure 11b groups instance by the suggested category by vendors. We can observe that most categories show a similar pattern except ones in the accelerated computing which is presented with a solid line with round markers. It is understandable situation that recent popularity of deep neural net, which requires significant computing power in the accelerated computing category, caused such lack of idle resource, which is also presented by Lee et. al. [28].

## B FURTHER ANALYSIS OF SPS PREDICTION MODEL

We demonstrated that AWS SPS dataset is predictable with decent accuracy and recall. In the SPS prediction, we applied multiple models and present the overhead of training and inference in Table 2. When measuring the time, we used SPS dataset gathered between July 24th. and July 31st. 2023. For inference, we used dataset from August 1st. to August. 3rd. In the time measurement, we used a dataset from a single instance type for comparison. We can observe quite difference in training time due to the model's complexity, especially Transformer [48]. We tried various Transformer optimization heuristics. However, considering the huge training overhead and lower prediction quality, it does not seem to be an appropriate approach to deal with time-series dataset, which coincides with the result presented by Elsayed et. al. [15]. The Prophet shows a very short training time with slightly lower prediction quality comparing to XGBoost as presented in Table 1. The train time difference is owing to the fact that XGBoost and other machine learning classifiers build separate models for each different output, which is 432. Meanwhile, Prophet builds a single model based on statistics of train dataset, and the train time did not get impacts from the output time window size. Considering the train and inference can happen off-line, we select XGBoost for the evaluation which shows better prediction quality.

## C FURTHER ANALYSIS OF THE SPOT INSTANCE AVAILABILITY EXPERIMENTS

Figure 12 presents a distribution of spot instance running time after fulfillment categorized by different dataset. The vertical axis shows the distribution, and the horizontal axis shows a running time without interruption in log-scale. The larger x-axis value

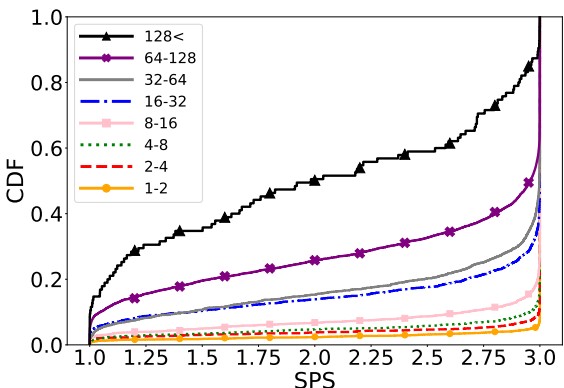

**(a) Categorization by CPU core numbers**

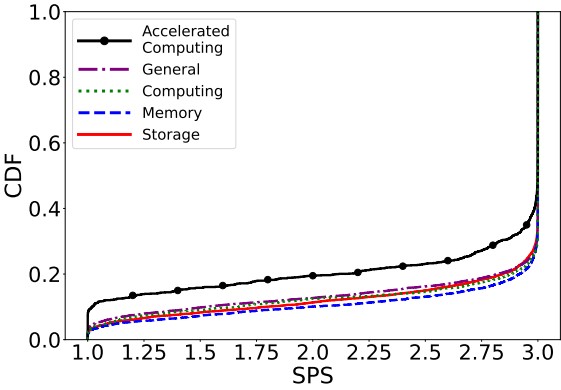

**(b) Categorization by Instance Category**

**Figure 11: SPS score distribution categorized by different criteria. SPS values show a noticeable distribution differences.**

| Model | Train Time | Inference Time |
|---|---|---|
| LR | 46 | 0.7 |
| RF | 75 | 0.9 |
| XGBoost | 806 | 1.2 |
| SVC | 626 | 15 |
| Prophet | 0.05 | 0.002 |
| ARIMA | 302 | 0.17 |
| Transformer | 2384 | 4.7 |

**Table 2: Train and Inference Time (seconds) of a SPS prediction model**

(to the right-side) implies more availability. Different sub-figures groups instances to different criteria; cost savings ratio (Figure 12a), interrupt-free score (Figure 12b), and instant availability (SPS) score (Figure 12c). In each figure, the yellow-solid, blue-dotted, and green-dashed lines indicate the result from AWS, Azure, and GCP, respectively. In each line, we mark a symbol of △ for *High* and ▽ for *Low* value after categorization.

From the figure, we can observe that the cost savings ratio (Figure 12a) does not show noticeable pattern between *High* and *Low*

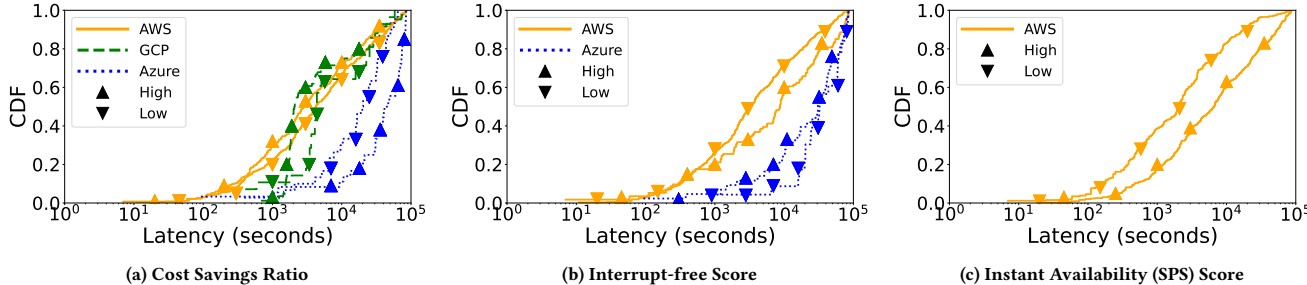

**(a) Cost Savings Ratio**                    **(b) Interrupt-free Score**                    **(c) Instant Availability (SPS) Score**

**Figure 12: A CDF of time until an interruption event happens for AWS, Azure, and GCP spot instances categorized by different dataset values of High (△) and Low (▽). A graph in the right side means higher availability.**

value for the spot instance running time. This observation concurs with the previous work that the price of the spot instance is not a good indicator for spot instance reliability [6, 21]. We can also observe that Azure shows the longest running time than the other vendors followed by GCP and AWS. For the interrupt-free score (Figure 12b), the AWS spot instance shows a noticeable pattern that

the higher interruption-free score is more reliable than the lower ones. However, the Azure does not show such a pattern. For the instant availability dataset (Figure 12c), which is provided by AWS only (SPS), instances with high scores show much more reliability than lower ones. On average, spot instances with the score of 3 runs for 4.7 hours, while that of score 1 runs only for 1.8 hours.