# OpenReview forum: "Making Cloud Spot Instance Interruption Events Visible"
_ACM.org/TheWebConf/2024/Conference — TheWebConf24 Oral_

### Official Review · Reviewer_6iDH · 2023-11-21

**Novelty:** 5
**Technical Quality:** 5

**Review:**

n this paper, the authors take an investigative look into the spot instance availability and reliability of AWS, Azure and GCP cloud providers. The authors conduct this analysis in two steps. First, they perform passive analysis on the datasets released by the cloud providers and attempt to see correlation between availability scores and other potentially affecting patterns (e.g. Time of day). The authors further conduct real-world experiements by reserving spot instances in all three providers global network until they observe interupption events. Based on the findings from both analysis, the paper also investigates if predicting future scores released by the providers can improve the spot instance selction resulting in more longevity. Overall, the paper provides several interesting insights for developers/researchers aiming to utilize spot instances and is highly relevant to the Web community.

## Strengths

- The paper tackles a super interesting problem which is more relevant after the policy changes of the cloud providers
- The paper is well written and easy to follow
- Some of the takeaways are super relevant and well-motivated

## Weakness

- This reviewer found section 3 of the paper particularly weak and hand-wavy in reaching the final conclusion. Specifically, the “time frame” of interruption dataset of AWS and Azure is not mentioned and it is not clear if the two times correlated with each other. Further, it is possible that the spot instance availability and interruptions are more correlated with instance type (more capable instances are of fairly limited availability in first place and therefore may get more interruptions as their demand increases) or dependent on geographical regions (US and EU may see more user traffic and usage compared to South America or Africa and therefore instances their might not get interrupted so often). However, for most of the analysis, the authors combine all instances and geographical values together and it is hard to understand the nuanced differences. The core analysis in the appendix gives some insight but it is also combining different cloud providers together.
- Section 3 also over-estimates the findings and takeaways, which may not encompass the reality. For instance, in Fig 3, the authors draw the conclusion on the daily and weekly patterns from a 1000th order difference in the SPS values. The SPS values in section 3 remain relatively stable and therefore the need of prediction is not well motivated here. Also the SPS conclusions shown in Fig 2 does not correlate with author’s own experiments as “On average, spot instances with the score of 3 runs for 4.7 hours, while that of score 1 runs only for 1.8 hours.” (In appendix) but the SPS values mostly hovers around 2.6 in section 3. The authors must clarify the takeaways by also showing the median and standard devision of the SPS values.
- The paper has limited exploration of GCP in sec 3 due to dataset availability which limits the correlations throughout the paper. The authors are advised to adjust the contributions of the paper to reflect this accurately.
- While the reviewer appreciated the real-world experiments and takeaways paper draws from correlating with the provider datasets, the details on the measurement methodology are lacking. This makes it difficult for future researchers to improve upon or replicate the work conducted by the authors in the paper.
- “Even after gathering the interruption event experiment results, without knowing the internal mechanism of spot instance operations, it can be challenging to select features that really impact the interruption events” → The paper also makes no attempt to shed light on the “internal mechanism” of spot operations. Please tone down this motivation in Intro.

**Questions:**

In addition to the weakness section above, the authors are requested to answer the following questions.

- How will the approach overcome future policy/dataset changes by AWS? What aspects from previous research in this domain can be leveraged as it is unlikely that cloud providers will change the spot availability schemes even if they change the scores/information in the dataset?
- Are the authors planning to release their experiment scripts to public?

**Reviewer Confidence:**

4: The reviewer is certain that the evaluation is correct and very familiar with the relevant literature

**Scope:**

4: The work is relevant to the Web and to the track, and is of broad interest to the community

---

### Official Review · Reviewer_GyoK · 2023-11-23

**Novelty:** 5
**Technical Quality:** 5

**Review:**

This paper analyzes various datasets of the spot instance and present the feasibility for value prediction. Then, to measure how the public datasets reflect real-world spot instance interruption events, the authors conduct real-world experiments for spot instances of AWS, Azure, and Google Cloud.

The problem is relevant. The paper is well written. The paper provides an interesting view of the current landscaper of cloud computing. It can ignite future research, although the specific contribution of this paper is relatively limited.

**Questions:**

- How can future researchers benefit from this analysis?
- If the current auction mechanism changes, would the result/methodology of the paper be still valid?

**Reviewer Confidence:**

3: The reviewer is confident but not certain that the evaluation is correct

**Scope:**

3: The work is somewhat relevant to the Web and to the track, and is of narrow interest to a sub-community

---

### Official Review · Reviewer_eQiU · 2023-11-23

**Novelty:** 5
**Technical Quality:** 5

**Review:**

Significance/Pros:

This topic is very interesting considering the significance of spot instance usage, given its broad interest. It's crucial for users employing such services to have a clear understanding. Clear structure and experimental details are clearly described.

Utilising data obtained from public cloud providers so the results are not based on the simulated data/information.

Offers a diverse range of relevant literature.

Quality: In general, the paper is well-written and clearly articulated, yet certain aspects require additional clarification.

Cons:

It's unclear whether the authors utilised datasets from SpotLake – which I assume yes?

Adding explicit details about the datasets used, clearly indicating their sources, would be beneficial. There are some repetitive statements.

Some relevant technical details and insights are missing which are listed as questions.

There are some repetitive statements.

Questions which needs further clarification:

1.	Were there instances where the ratio of savings to interruptions was higher compared to other instance types? This could provide meaningful insights. How did you compare instances of different types across various CPs?
2.	Figure 2a - it is apparent that AWS has the least cost savings. Is it same across all the instance types?
3.	Could you provide more information about the dataset's size and specifics? How many instance categories are there, along with their regions and entries?
4.	Figure 2b presents the temporal change pattern - Is it consistent across all regions that offer the same instance type? Have you noticed any regional impact on the pattern?
5.	SPS score/ time when cloud usage is low - Is this specific to a particular region or consistent across all regions?
6.	How do you compare your work/RF-based findings to those mentioned in Reference 29?
7.	On AWS, the SPS score ranges from 1 to 10. How does this mapping align with equation 1?
8.	What is the rationale behind the mapping of dataset with values in the range of five categories to a numeric value between 1.0 and 3.0 increments by 0.5?
9.	What's the reasoning behind choosing these algorithms, given that some are tailored to time series while others are generic classifiers?
10.	Their accuracies are nearly identical — is there an underlying aspect that would lead you to prefer one over another? Also, it is not convincing that XGBoost is performing the best of all.
11.	The author stated an of 63.2% and 168% increase in the spot instance running time, could this be aided with additional clarification when compared to the base model or findings?

**Questions:**

Questions which needs further clarification:

1.	Were there instances where the ratio of savings to interruptions was higher compared to other instance types? This could provide meaningful insights. How did you compare instances of different types across various CPs?
2.	Figure 2a - it is apparent that AWS has the least cost savings. Is it same across all the instance types?
3.	Could you provide more information about the dataset's size and specifics? How many instance categories are there, along with their regions and entries?
4.	Figure 2b presents the temporal change pattern - Is it consistent across all regions that offer the same instance type? Have you noticed any regional impact on the pattern?
5.	SPS score/ time when cloud usage is low - Is this specific to a particular region or consistent across all regions?
6.	How do you compare your work/RF-based findings to those mentioned in Reference 29?
7.	On AWS, the SPS score ranges from 1 to 10. How does this mapping align with equation 1?
8.	What is the rationale behind the mapping of dataset with values in the range of five categories to a numeric value between 1.0 and 3.0 increments by 0.5?
9.	What's the reasoning behind choosing these algorithms, given that some are tailored to time series while others are generic classifiers?
10.	Their accuracies are nearly identical — is there an underlying aspect that would lead you to prefer one over another? Also, it is not convincing that XGBoost is performing the best of all.
11.	The author stated an of 63.2% and 168% increase in the spot instance running time, could this be aided with additional clarification when compared to the base model or findings?

**Reviewer Confidence:**

4: The reviewer is certain that the evaluation is correct and very familiar with the relevant literature

**Scope:**

4: The work is relevant to the Web and to the track, and is of broad interest to the community

---

### Official Review · Reviewer_RFsa · 2023-11-24

**Novelty:** 3
**Technical Quality:** 4

**Review:**

This paper explores the reliability of spot instances by statistically analyzing availability datasets of preemptible instances of major public cloud computing vendors. The authors evaluated several existing predictors on the AWS Spot Placement Score (SPS) dataset. This paper also uses the Kaplan-Meier estimator to model the probability of spot instance survival rate. Finally, the authors concluded this paper with some research insights to guide readers to utilize cloud resources more efficiently.

Pros:

(1)	This paper studies the reliability of spot instances, which might be an important and well-studied problem for cloud-computing-powered companies.

(2)	This paper provides rich line charts to explain the statistical and distributional features of studied datasets.

(3)	This paper includes data from major cloud computing vendors such as AWS, Azure, and Google Cloud Platform (GCP), which are very representative of the current cloud computing market.

Cons:

(1)	The novelty of this paper is not enough. Section 3 "SPOT INSTANCE DATASET CHARACTERIZATION" contains just basic data analysis. Section 4 "PREDICTING SPOT INSTANCE DATASET CHANGE" evaluates existing prediction methods. Subsection 5.1 "Spot Instance Interrupt Analysis" uses a well-established statistical model.

(2)	This paper is not presented in review mode, which does not meet the recommended formatting requirements of the conference.

(3)	The settings of SPS prediction experiments in this paper are not reasonable, and the evaluated results of these experiments are almost meaningless. In section 4 "PREDICTING SPOT INSTANCE DATASET CHANGE" and subsection 5.2 " Effectiveness of Instant Availability Dataset", the SPS data are categorized in a bin size of 1 and 0.5, respectively. However, as shown in Figure 2(c), the values of SPS data nearly stay at 2.66 and could be always binned to 3. In this way, the important fluctuation in the original data is totally neglected and the prediction task becomes to predict a nearly constant discrete series which is very easy. As a consequence, as shown in Table 1 and Figure 5, the prediction accuracy is always very high for every model and in every setting.

(4)	The meaning of the SPS metric is not clearly introduced in the main paper. SPS is used throughout the whole paper but is only further studied in Appendix A.

(5)	There are certain writing issues in this paper. For example, in section 6 "RELATED WORK", there are two titles "Characterizing Spot Instance Dataset:" and "Modeling and Using Spot Instance Dataset:" for the first paragraph.

(6)	In Figure 2, the scales of the Y-axes of three sub-figures are inconsistent and unreasonable. For example, in sub-figure "(b) Interruption-Free Score Average", the trend of the change of AWS through time is very hard to tell.

**Questions:**

(1)	Concerning the prediction experiments in section 4 "PREDICTING SPOT INSTANCE DATASET CHANGE", why the accuracy results are so close across every setting?

(2)	In Figure 6, what does the X-axis " Feature (Time Recency)" mean? And why the most important features are in the negative value range (roughly around minus 20)?

**Reviewer Confidence:**

3: The reviewer is confident but not certain that the evaluation is correct

**Scope:**

3: The work is somewhat relevant to the Web and to the track, and is of narrow interest to a sub-community

---

### Decision · Program_Chairs · 2024-01-22

**Decision:**

Accept (Oral)

**Comment:**

This paper focuses on cloud spot instances and their availability and reliability. The topic is of relevance to this community. Reviewers have praised that the authors investigate data from the major 3 cloud providers. However, reviewers have raised the following as areas for improvement: clarity of the contributions, granularity of SPS bins, detail of some data and graphs. The authors' rebuttal is detailed, but they did not answer all concerns (e.g. the time frame issue from reviewer 6iDH). Overall, I recommend this paper to be accepted, but I will not argue strongly in favor.